# High Efficiency Crumpled Carbon Nanotube Heaters for Low Drift Hydrogen Sensing

**DOI:** 10.3390/s19183878

**Published:** 2019-09-09

**Authors:** Jeonhyeong Park, Il Ryu Jang, Kyungtaek Lee, Hoe Joon Kim

**Affiliations:** Department of Robotics Engineering, Daegu Gyeongbuk Institute of Science & Technology (DGIST), Daegu 42988, Korea (J.P.) (I.R.J.) (K.L.)

**Keywords:** carbon nanotubes, crumpled CNTs, joule heating, hydrogen sensing, gas sensing

## Abstract

This work presents the fabrication of crumpled carbon nanotubes (C-CNTs) thin film heaters and their application towards high sensitivity and low drift hydrogen gas sensing. Utilizing a spray coating of pristine multi-walled carbon nanotubes (MWCNTs) and thermal shrinkage of polystyrene (PS) substrate, we have fabricated C-CNTs with closely packed junctions. Joule heating of C-CNTs gives higher temperature at a given input voltage compared to as-deposited CNTs. In addition, temperature coefficient of resistance (TCR) is analyzed for accurate temperature control and measurement of the heater. The C-CNT heaters are capable of hydrogen gas sensing while demonstrating higher measurement sensitivities along with lower drift compared to as-deposited CNT devices. In addition, the self-heating of C-CNT heaters help rapid desorption of hydrogen, and thus allowing repetitive and stable sensor operation. Our findings reveal that both CNT morphologies and heating temperatures affect the hydrogen sensing performances.

## 1. Introduction

Carbon nanotubes (CNTs) gas sensors have drawn much attention due to their high sensitivity, simple fabrication process, and fast response time [1,2]. Out of several existing detection mechanisms, the chemiresistive sensing scheme is most widely used as it is power efficient and easy to adapt with existing electronic components [3,4]. Multi-walled CNTs-based chemiresistive sensors exhibit semiconducting properties along with high surface-to-volume ratio, and thus providing high measurement sensitivities [5,6]. Previous studies range from the development of single CNT wire integrated transistor-type sensors [7,8] to flexible sensors for wearable applications [9]. In addition, economical methods of CNTs integration, such as direct-growth, spray coating, transfer printing, and spin coating, have allowed the scalable production of CNT-based gas sensors [10,11,12]. Although CNT gas sensors possess numerous aforementioned advantages, such sensors leave a lot to be desired in terms of the measurement reliability or ability to make repetitive measurements [13,14].

In the chemiresistive sensing scheme, the adsorption or adhesion of gas molecules onto CNTs result in shift in its electrical resistance [1,15]. To make reliable and repetitive measurements, the adsorbed gas molecules need to be removed from CNTs when the gas is not present. At room temperature, desorption takes a long time and even results in a permanent shift in sensors electrical resistance [16,17]. Such drift in electrical resistance is undesired, especially in applications where one needs to make repetitive and lengthy measurements. Utilizing external energy sources, such as UV light induced photo desorption, high vacuum operation, or thermal annealing can expedite the desorption process [18]. Due to its simplicity, the thermal annealing method is widely used to minimize the drift of CNTs-based gas sensors. For example, annealing the CNT hydrogen gas sensors at near 100 °C allows the rapid recovery of sensor resistance [19]. 

CNTs-based joule heaters exhibit reliable thermal operations and can be patterned on flexible substrates. The majority of heat generates at CNT junctions and the amount of CNTs determines the electro-thermal properties of the heaters [20,21,22]. Since a layer of CNTs itself can work as a joule heater, such platform is ideal for gas sensing applications at elevated temperatures. However, to the best of our knowledge, there is a lack of studies, which utilize both self-heating functions and gas sensing abilities of CNTs. As previously mentioned, the heating properties of CNT heaters rely on the amount of CNT junctions in a given volume as the majority of heating occurs at the junctions. Higher junction densities should lead to more power efficient heating and compressing or crumpling CNTs can drastically increase such junction densities.

Here, we report the fabrication of C-CNT heaters using a spray coating and thermal shrinkage method followed by the electrical and thermal property characterization of the heater. Using the C-CNT heater, we detect hydrogen gas and analyze the measurement sensitivity, drift in sensor resistance, sensing repeatability and reliability, and temperature dependent sensor operation. We compare the performances of C-CNT sensors to as-deposited CNT platform to highlight the advantages of the proposed C-CNT platform.

## 2. Materials and Methods

### 2.1. Fabrication of Crumpled CNT Sensor

CNT thin film heaters rely on localized heat generation at CNT junctions when current flows through CNT networks [23]. High junction resistances induce rapid increase in temperature. The proposed device utilizes such joule heating effect and consists of a CNT layer on top of a 200-μm-thick polymer substrate and metal electrodes, as shown in Figure 1a. To make direct comparison between conventional as-deposited CNT heaters and C-CNT heaters, we have used two different substrates of polyethylene terephthalate (PET) and polystyrene (PS). PET is thermally resistant while PS shrinks about 50% in length when heated up to 150 °C [24]. The initial amount and deposition area of CNTs are same for both PET and PS substrates, but the final sizes are different after the thermal shrinkage process.

Figure 1b depicts the overall fabrication process of the C-CNT heater. A 3 wt% MWCNT solution is diluted in isopropyl alcohol (IPA) at 0.1 wt% and then spray coated for 10 s at the flow rate of about 0.4 mL/s. The heater is then rinsed with IPA for cleaning. Following the cleaning process, the heater is placed inside the convection oven at 70 °C for 30 min to remove any IPA vapors remaining in CNTs. Such thermal annealing step also enhances the adhesion between CNTs and the polymer substrate. To crumple CNTs, the device fabricated on PS substrate is heated at the temperature of about 150 °C for 5 min using the convection oven. The heater size reduces by about 400% in area after the thermal shrinkage process. Finally, a layer of silver paste followed by a 100-nm-thick metal layer (Au or Pt) is sputter deposited at the sides of device to form electrodes.

### 2.2. Device Characterization and Hydrogen Sensing Setup

For an accurate temperature control and measurement, the electro-thermal properties of as-deposited CNT and C-CNT heaters are characterized. MWCNTs show linear temperature coefficient of resistance (TCR) [25], hence its temperature can be calibrated from the measured electrical resistance. TCR of fabricated heaters are measured while monitoring the heater temperature using an infrared (IR) camera with varying input voltage. To accurately measure the heater resistance (*R*_heater_), a circuit consisting of DC power supply, a 100 Ω sense resistor (*R*_sense_), and the CNT heater is used, as shown in Figure 2. Such method allows the measurement of *R*_heater_ by calibrating the voltage drop across *R*_sense_ (Equation (1)). All device characterizations are carried out at the room temperature of about 20 °C in a humidity-controlled environment.
*R*_heater_ = *R*_sense_ × (*V*_in_ − *V*_sense_)/*V*_sense_(1)

Figure 2 shows the experiment setup for hydrogen gas sensing. The experiment is performed inside the vacuum probe station with accesses to argon and hydrogen gases. The gas flow rate of 500 sccm with 10% hydrogen concentration (Ar 450 sccm, H_2_ 50 sccm) is used throughout the sensing experiment except the sensor characterization as a function of varying hydrogen concentrations. The chamber pressure is maintained at 400 mT using the automatic purge valve control. To minimize any conduction heat loss, we have placed the CNT sensors about 0.5 cm above the surface. Hydrogen gas is introduced after 7 min for both pressure and sensor temperature stabilization throughout the entire sensing experiments. Each sensing cycle is 8 min long while hydrogen is introduced for 4 min and then turned off for another 4 min. During the hydrogen sensing experiment, *R*_heater_ is recorded every 100 ms as hydrogen adsorption would alter *R*_heater_ via electron-hole recombination [26].

## 3. Results

### 3.1. Device Characterization

Figure 3a shows the fabricated as-deposited CNT heater on a PET substrate and C-CNT heater on a PS substrate. The C-CNT device size has reduced by 400% after the thermal shrinkage process. Although the initial area and amount of deposited CNTs are identical for both heaters, the C-CNT heater appears to be darker indicating that the areal density of CNTs have increase after the shrinkage. The measured resistances of CNT and C-CNT heaters at room temperature are 12.4 kΩ and 5.1 kΩ, respectively. For demonstration, we have fabricated a larger CNT heater on a PET substrate and attached to the beaker filled with water, as shown in Figure 3b. Figure 3c shows the conformal temperature distribution of the heater measured using an IR camera. Both CNT and C-CNT heaters have shown stable thermal operation of up to 80 °C without any mechanical deformation or delamination of the CNT layer.

Figure 4a,b shows scanning electron microscope (SEM) micrographs of as-deposited CNT and C-CNTs. Spray coated CNTs are randomly entangled while C-CNTs show much denser bundles of entangled CNTs due to a reduction in area of the PS substrate [27,28]. In addition, there is more CNT junctions in a given area for C-CNTs, which leads to more efficient heating due to less heat loss. Although the morphology of CNTs has changed through the thermal shrinkage process, Raman analysis confirms that the CNT quality did not degrade, as shown in Figure 4c. For both as-deposited CNTs and C-CNTs, typical Raman peaks of CNTs (D-peak, G-peak, and 2D-peak) are present [29] with fairly consistent G-peak to D-peak ratio. In case of as-deposited CNT samples, the PET Raman peak is present since the Raman laser can reach the PET substrate. In contrast, there is no PS peaks for C-CNT samples as the transparency decreases with increased CNT densities per given area due to the thermal shrinkage process. This indicates that the proposed thermal shrinkage method is reliable and does not induce chemical transformations of the material.

Figure 5 shows the electro-thermal property characterization of the fabricated devices. Since C-CNTs have higher junction densities, it demonstrates better heating efficiencies compared to as-deposited CNTs. At a given voltage or power, the C-CNT device exhibits higher temperature than the CNT device, as shown in Figure 5a,b. Such high heating efficiency of the C-CNT device is a great advantage for gas sensing applications as the high temperature environment is favorable not only for enhanced desorption of gas molecules but also for the adsorption rate according to thermodynamics [14]. Figure 5c shows change in *R*_heater_ as a function of temperature. For as-deposited CNTs, TCR is measured to be −892.0 ppm/°C, which is close to the previously reported value for MWCNTs [19]. In contrast, the TCR of C-CNT is lower at −590.5 ppm/°C. MWCNTs possess both metallic and semiconducting tubes. A compressive stress induced from thermal shrinkage process can change the chirality from semiconducting to metallic tubes [30], and thus ultimately affecting TCR of CNTs. Although TCR value shifted after the shrinkage, both CNT and C-CNT heaters exhibit highly linear relation between *R*_heater_ and the heater temperature. Such linear TCR is favorable for accurate temperature measurement and control of the device. Figure 5d shows a linear I-V relation meaning the change in *R*_heater_ is rather small while the devices are heated.

### 3.2. Hydrogen Gas Sensing

The fabricated CNT heaters can work as sensors to measure hydrogen gas. Figure 6 shows the response of both as-deposited CNT and C-CNT sensors to 10% concentration of hydrogen gas at ambient temperature of about 20 °C. Here, the sensitivity is defined using the Equation (2), where *R*_0_ is the initial sensor resistance before introducing hydrogen. It is clear that the C-CNT sensor gives higher sensitivity along with more stable operation than the as-deposited CNT sensor. However, the measured sensitivity values are in the 0.01% range and exhibits severe drift due to poor adsorption and desorption rates of hydrogen at room temperature.
Sensitivity = (*R*_heater_ − *R*_0_)/*R*_0_ ×100 [%](2)

To improve adsorption and desorption rates of hydrogen, we have performed the gas sensing experiment while both as-deposited CNT and C-CNTs sensors are self-heated. Figure 7a,b shows the change in sensitivities at input voltages of 10 V and 20 V. Even at same input voltages, the heating temperatures are higher for C-CNT sensors owing to its high heating efficiencies. Such higher sensing temperatures lead to an increase in hydrogen adsorption rates, and thus enable high sensitive hydrogen detection. For example, C-CNT sensors exhibits about 350% improvement in measurement sensitivity compared to as-deposited CNT sensors at 20 V of input voltage. Compared to the room temperature operation, the measurement sensitivities have drastically improved at elevated sensor temperatures, showing that the proposed self-heating mechanism is relatively simple but very effective for high sensitivity hydrogen gas sensing.

In addition, the drift drastically reduced at elevated temperatures, as shown in Figure 7d. Equation (3) provides the drift factor, where *R*_n_ is the *R*_heater_ at the beginning of each sensing cycle and Δ*R* is the amount of rise in *R*_heater_ when hydrogen gas is present. Using the drift factor, we can calculate how much *R*_heater_ is drifting as the sensing cycle continues. The drift factor decreases for both as-deposited CNT and C-CNT sensors as the input voltage increases, which clearly shows that the self-heating mechanism lowers the sensor drift due to enhanced the hydrogen desorption.
Drift Factor = (*R*_(n+1)_ − *R*_n_)/∆*R* × 100 [%](3)

Figure 8 shows the response of the as-deposited CNT and C-CNT sensors to different concentrations of hydrogen gases at an input voltage of 20 V. For both types of sensors, sensitivities increase linearly with hydrogen concentration. The rate of change is about 2.6 times higher for C-CNT sensors along with better linear dependency. Such results suggest the proposed C-CNT sensor is capable of not only detecting the presence of hydrogen but also for sensing the gas concentration. Both response time and recovery time of the sensors remained consistent for varying hydrogen concentrations. In addition, we have analyzed the limit of detection (LOD) of the developed CNT sensors using the root-mean-square deviation (rmsd) of the baseline signal, *R*_rms_ [16]. For rmsd calculations, we have used 100 data points and the LOD can be acquired using the below equation, which considers the signal-to-noise ratio larger than 3 as meaningful signals.
LOD = 3 × *R*_rms_/Slope [ppm](4)

The calculated LOD from the measurements are 7100 ppm and 2700 ppm for CNT and C-CNT sensors, respectively. Although LOD values are higher than that of the functionalized CNT-based hydrogen sensors, it is still comparable to a previously reported LOD value [31]. Moreover, the C-CNT sensor exhibits a 260% improvement in LOD compared to as-deposited CNT sensors.

To analyze the repeatability of the C-CNT sensor, we have analyzed its sensing performances for extended hydrogen sensing cycles. Figure 9a shows the response of as-deposited CNT and C-CNT sensors for 20 cycles, which corresponds to about 3 hours, of hydrogen sensing at 10% concentration. The overall shift in sensitivity is consistent throughout measurement with small drift via self-heating of the sensors. Such consistent sensor performance shows that the C-CNT sensors can be used for extended periods to make repeatable gas sensing measurements. In addition, we have fabricated at least 4 sensors of each design and they have demonstrated similar heating and sensing performances, suggesting that the presented device fabrication is highly reproducible. Figure 9b,c compares the heating characteristics of as-deposited CNT and C-CNT sensors before and after the 20 cycles of hydrogen gas sensing. Both TCR and power-to-sensor temperature relations remain fairly constant, indicating that our sensors show high stability in electro-thermal properties. 

## 4. Discussion

The presented C-CNT sensors utilize high CNT junction densities for efficient heating. In addition, a self-heating of the sensor improves the measurement sensitivity and alleviates the sensor drift in hydrogen gas sensing. Table 1 compares the performances of the developed sensors to selected previous works on CNT-based hydrogen gas sensors. To the best of our knowledge, this work presents the utilization of crumpled CNT for the first time. Although the measurement sensitivity and LOD of C-CNT sensors are somewhat limited compared to functionalized CNT sensors, we believe a proper functionalization process along with the design optimization could further enhance our device performance. In addition, many of previous works implement external heating methods to improve the device sensitivity while our approach uses the sensor itself as the only heat source.

When comparing the as-deposited CNT and C-CNT sensors, the measurement sensitivity is similar at a given heated temperature for as-deposited CNT and C-CNT sensors. However, C-CNT sensors exhibit more stable drift performance along with improved heating efficiencies, which is preferable for low-power or low-voltage applications. Even without the self-heating of sensors, C-CNT sensors still demonstrate much more reliable sensor operation along with and higher measurement sensitivities. Such better sensing performances without heating showcase that the proposed C-CNT structures could be used to develop high performance room temperature gas sensors for single detection applications [13].

Like other existing CNT-based chemiresistive sensors, several aspects are needed to be considered for an accurate and reliable sensor operation. Since *R*_heater_ depends on sensor temperature as well as the amount of hydrogen, it is necessary to compensate and properly account for the temperature effect from environments. For example, 1 °C change in environmental temperature can induce 0.059% shift in *R*_heater_, which could be significant for pristine CNT-based chemical sensors. In addition, the electrical resistance of CNTs also depends on humidity [35] and other types of gases, such as CO, NO_2_, CH_4_, and O_2_ [1]. Temperature or humidity issues can be addressed by integrating calibration sensors while the gas selectivity problems can be mitigated via functionalization with chemical agents [36,37,38,39,40,41]. Moreover, the proposed C-CNT sensor improves adsorption and desorption rates compared to as-deposited CNT sensors without any chemical treatment, and we believe the sensor performance would further improve via the proper functionalization process to detect not only hydrogen, but also for other chemicals.

Another way to improve the sensitivity, response time, and recovery time is to scale down the sensor size [42]. Since the proposed thermal shrinkage process reduces the CNT area by 400% from the original size, even μm length scale C-CNT sensors can be realized by using a shadow mask with sub-mm feature sizes. The presented fabrication approach is scalable and could allow a batch fabrication of C-CNT sensor arrays. In addition, higher self-heating temperatures would lead to enhanced sensitivities and drift performances based on our findings. A fabrication or transfer printing of C-CNT sensors onto thermally resistant substrates (glass, Si, etc.) could lead to higher self-heated temperatures, which would result in improved sensor performances. All aforementioned approaches could further improve the current state of the proposed C-CNT sensor, which exhibits superior heating efficiency, gas detection sensitivity and repeatability, and less drift under cyclic sensing of hydrogen gas compared to as-deposited CNT sensors.

## 5. Conclusions

This work proposes crumpled CNT (C-CNT) sensors with efficient self-heating abilities for high sensitivity and low drift hydrogen gas sensing. The fabrication process implements a rather simple spray coating to deposit a conformal layer of CNTs. In addition, a thermal shrinkage of PS substrate enables a reliable reproduction of C-CNT structures with high junction densities in a given area. The C-CNT sensors are more voltage and power efficient compared to as-deposited CNT devices. In addition, a highly linear TCR allows accurate temperature measurement and control. In ambient conditions, C-CNT sensors exhibit higher response to hydrogen gas. Moreover, elevated sensing temperatures via self-heating of the sensors increase both hydrogen adsorption and desorption rates, which ultimately improved the sensor performances while maintaining outstanding sensing repeatability and stability. Since the proposed C-CNT structures are more responsive to hydrogen and possess high heating efficiencies compared to the conventional as-deposited CNT sensors, C-CNT platforms could be applied towards an ultra-low power and high sensitivity hydrogen gas detection. In addition, the sensor itself works as a thermistor, and thus the environment temperature fluctuations can be addressed when necessary. With a proper chemical functionalization and improved device design, we envision the application of C-CNT sensors to other chemical sensing applications beyond hydrogen gas detection.

## Figures and Tables

**Figure 1 sensors-19-03878-f001:**
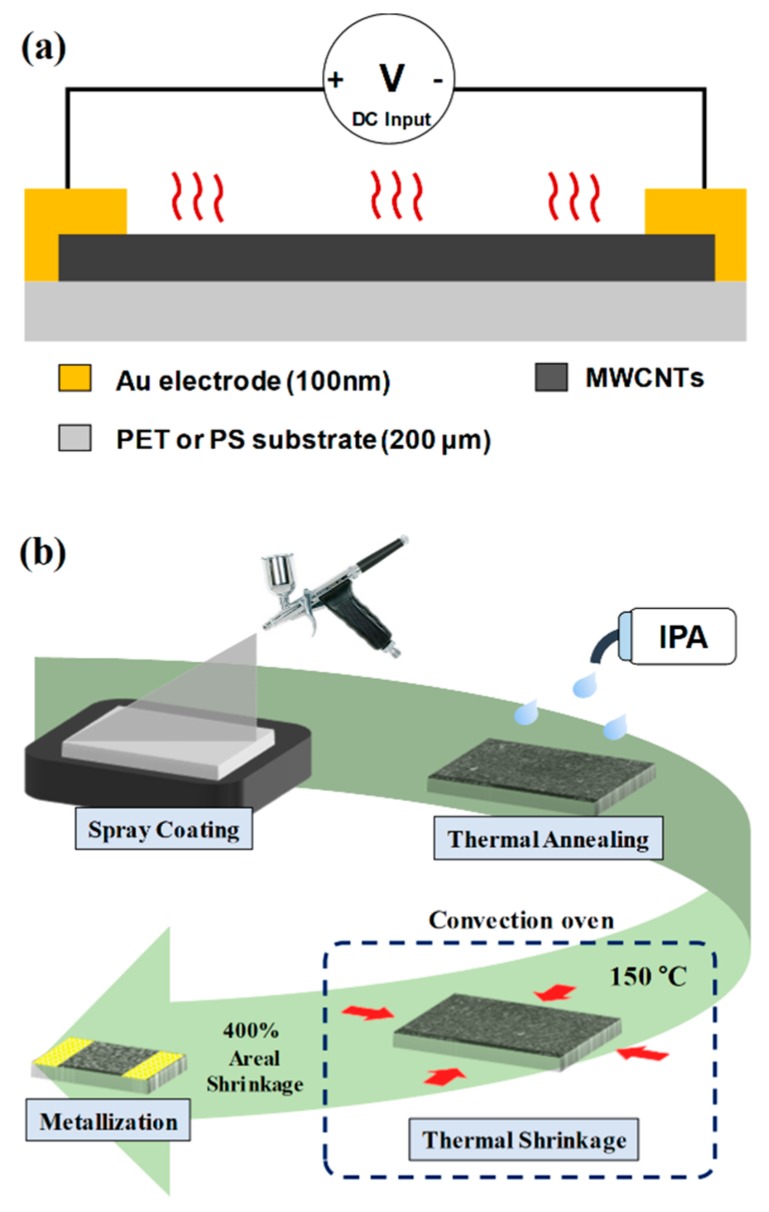
(**a**) A schematic of the crumpled carbon nanotube (C-CNT) heater. (**b**) Fabrication process including a CNT spray coating, a thermal annealing, a shrinkage of a polystyrene (PS) substrate, and the metallization step.

**Figure 2 sensors-19-03878-f002:**
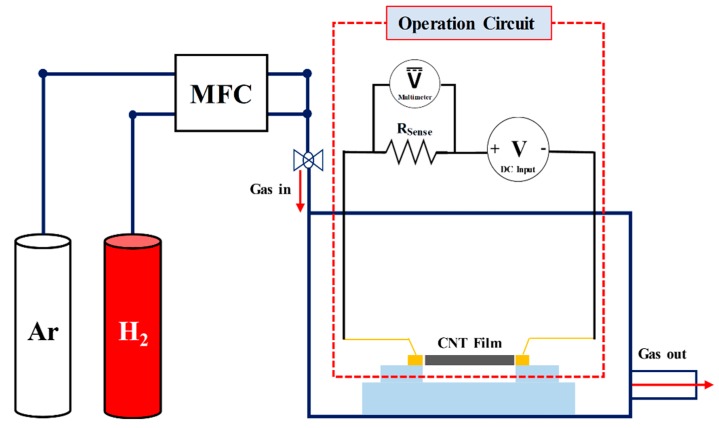
Experiment setup for a hydrogen gas sensing and CNT heater operation. By measuring the voltage drop (*V*_sense_) across the 100 Ω sense resistor (*R*_sense_), the sensor resistance can be calibrated.

**Figure 3 sensors-19-03878-f003:**
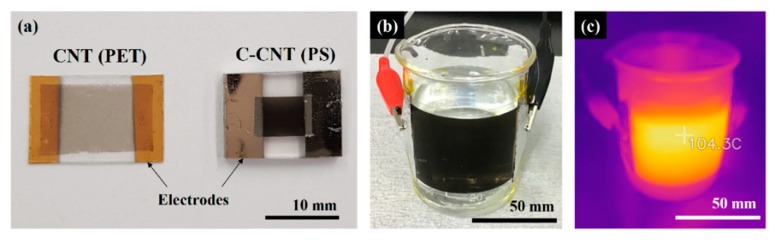
(**a**) As-deposited CNT heater on a polyethylene terephthalate (PET) substrate and C-CNT heater on a thermally shrunk PS substrate. Thermal shrinkage induces about 400% decrease in heater size. (**b**) Application of an as-deposited CNT heater on a curved surface. (**c**) Infrared (IR) image of the heated CNT heater.

**Figure 4 sensors-19-03878-f004:**
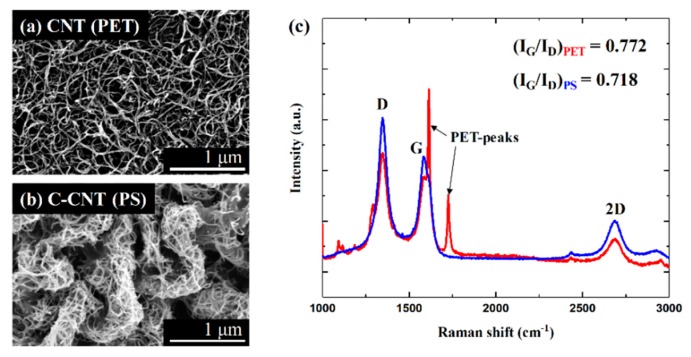
SEM micrographs of (**a**) as-deposited CNT and (**b**) C-CNT. (**c**) Raman spectra of as-deposited CNT and C-CNT confirms that the thermal shrinkage process does not affect the CNT.

**Figure 5 sensors-19-03878-f005:**
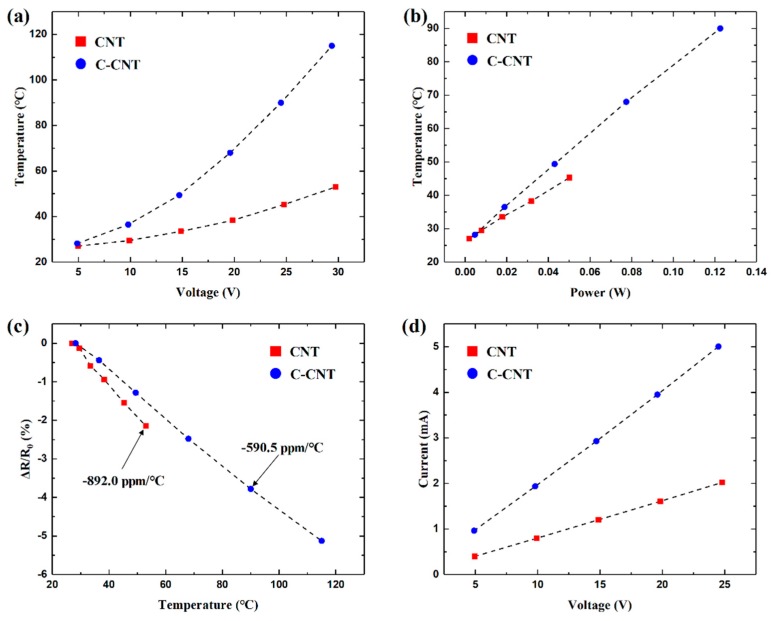
Measured sensor temperatures as functions of applied (**a**) DC voltage and (**b**) heater power. (**c**) Temperature coefficient of resistance as a function of the sensor temperature. (**d**) I-V measurement of the sensors.

**Figure 6 sensors-19-03878-f006:**
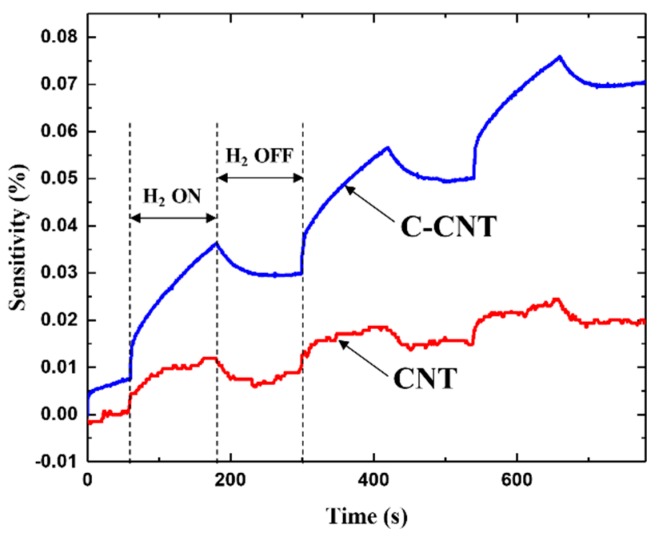
Hydrogen sensing responses of CNT and C-CNT sensors at room temperature of 20 °C. Although the C-CNT sensor shows higher sensitivity, a severe drift in sensor resistance is present.

**Figure 7 sensors-19-03878-f007:**
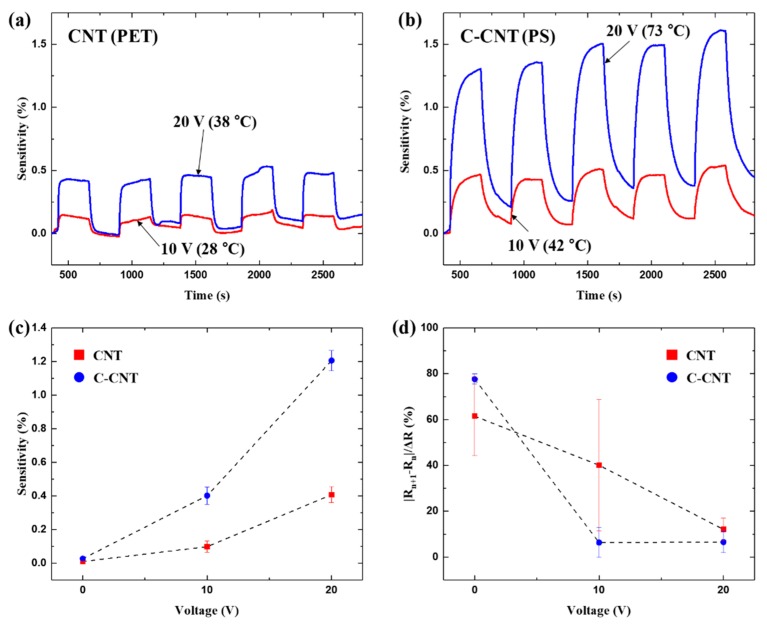
Hydrogen sensing responses with 10 V and 20 V input voltages for (**a**) CNT and (**b**) C-CNT sensors. (**c**) Measured sensitivities for the sensors as a function of applied voltage. C-CNT sensor shows much higher sensitivity at a given input voltage. (**d**) Drift in sensor resistance for CNT and C-CNT sensors. Drift decreases with increases sensing temperatures.

**Figure 8 sensors-19-03878-f008:**
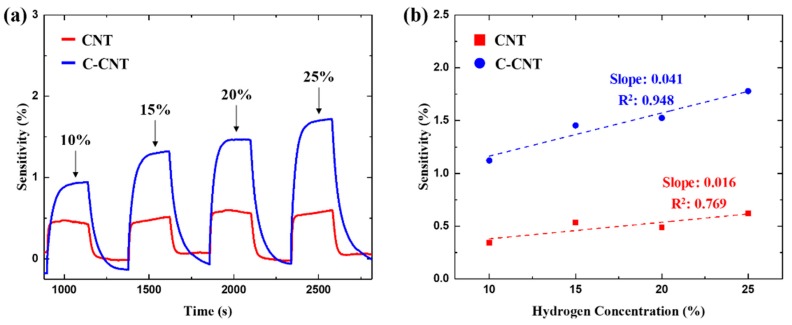
(**a**,**b**) Sensitivity of sensors (*V*_in_ = 20 V) at varying hydrogen concentrations. C-CNT sensors show a higher rate of change in measurement sensitivity compared to CNT sensors.

**Figure 9 sensors-19-03878-f009:**
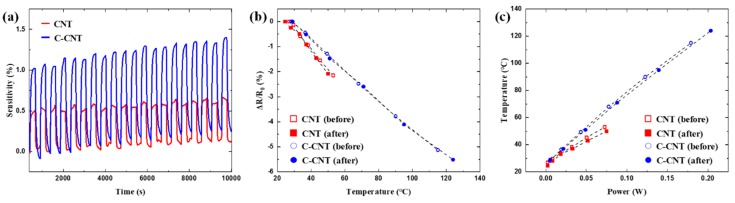
(**a**) Sensitivity of as-deposited CNT and C-CNT sensors with 20 V input voltage for 20 cycles of hydrogen sensing at 10% concentration. (**b**,**c**) Measured temperature coefficient of resistance (TCR) and power-to-sensor temperature relations before and after the hydrogen sensing experiments.

**Table 1 sensors-19-03878-t001:** Summary of CNT-based hydrogen gas sensors including the devices from this work.

CNT Type	Functionalization	Sensitivity	LOD	Heating	Reference
**MWCNT**	No	~0.4% (10% H_2_)	7100 ppm	Self-heated (20 V, 42 °C)	This work
**Crumpled MWCNT**	No	~1.3% (10% H_2_)	2700 ppm	Self-heated (20 V, 73 °C)	This work
**MWCNT**	No	~12% (18% H_2_)	N/A	External (100 °C)	[18]
**MWCNT**	Pd Pd/Pt	~4% (1% H_2_) ~2% (1% H_2_)	2000 ppm 400 ppm	No	[31]
**MWCNT**	MnO_2_	~15% (18% H_2_)	N/A	External (220 °C)	[32]
**MWCNT**	Pt/TiO_2_	~5% (20% H_2_)	N/A	External (50 °C)	[33]
**MWCNT**	Pt/f-GNPs	~17% (4% H_2_)	N/A	No	[34]

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
