# Peer review of "High Efficiency Crumpled Carbon Nanotube Heaters for Low Drift Hydrogen Sensing"

_sensors, 2019, doi:10.3390/s19183878_

Round 1

Reviewer 1 Report

Some comments should be answered before the consideration of publication.

There is no mention of the limit of detection of the sensor. No data on repeatability of the sensor response upon repeated cycles of gas exposure and pumping. No comment on the reproducibility, i.e., whether there were many devices tested showing similar behavior. There is no table to summarise the results of previous H2 sensors and to compare those with the current study without which it is not possible for the reader to realise the importance of this work. The following related references are recommended to be added in the manuscript.

[1] http://dx.doi.org/10.1155/2009/493904

[2] DOI: 10.1007/s00542-016-3154-2

[3] https://doi.org/10.1007/5346_2014_59

[4] IEEE Sensors Journal, vol. 13, no. 6, pp. 2423-2427, April 2013.

[5] DOI: 10.1007/s00542-018-3712-x

[6] ECS Journal of Solid State Science and Technology, vol. 6, no. 10, pp.   

     M130-M132, November 02, 2017.

Author Response

Dear Reviewer:

Best Regards,

Joon

Reviewer 2 Report

Fig.4 reveals the SEM images and Raman spectra of as-deposited CNT and C-CNTs. Why the morphology changed after the  thermal shrinkage process?Why the PET peaks are presented on the Raman spectra for the CNT sample and has disappeared after thermal shrinkage process. The stability experiments of heater characteristics should be carried out (for example, drift of TCR value and etc) Also the experiments  on heater characteristics drift after the heaters were exposed by H2 gas also should be done.

Author Response

Dear Reviewer:

Best Regards,

Joon

Round 2

Reviewer 1 Report

The manuscript has been revised well.

Reviewer 2 Report

am satisfied with the answer of the authors and I think that the article can be accepted for publication